# Joint Communications and Sensing Employing Multi- or Single-Carrier OFDM Communication Signals: A Tutorial on Sensing Methods, Recent Progress and a Novel Design

**DOI:** 10.3390/s22041613

**Published:** 2022-02-18

**Authors:** Kai Wu, Jian Andrew Zhang, Xiaojing Huang, Yingjie Jay Guo

**Affiliations:** Global Big Data Technology Centre (GBDTC), University of Technology Sydney (UTS), Sydney, NSW 2122, Australia; kai.wu@uts.edu.au (K.W.); andrew.zhang@uts.edu.au (J.A.Z.); xiaojing.huang@uts.edu.au (X.H.)

**Keywords:** joint communications and sensing (JCAS), Internet of Things (IoT), orthogonal frequency division multiplexing (OFDM), radar sensing, multi-carrier, single-carrier, discrete Fourier transform (DFT), fast Fourier transform (FFT), decimation

## Abstract

Joint communications and sensing (JCAS) has recently attracted extensive attention due to its potential in substantially improving the cost, energy and spectral efficiency of Internet of Things (IoT) systems that need both radio frequency functions. Given the wide applicability of orthogonal frequency division multiplexing (OFDM) in modern communications, OFDM sensing has become one of the major research topics of JCAS. To raise the awareness of some critical yet long-overlooked issues that restrict the OFDM sensing capability, a comprehensive overview of OFDM sensing is provided first in this paper, and then a tutorial on the issues is presented. Moreover, some recent research efforts for addressing the issues are reviewed, with interesting designs and results highlighted. In addition, the redundancy in OFDM sensing signals is unveiled, on which, a novel method is based and developed in order to remove the redundancy by introducing efficient signal decimation. Corroborated by analysis and simulation results, the new method further reduces the sensing complexity over one of the most efficient methods to date, with a minimal impact on the sensing performance.

## 1. Background and Motivation

Joint communications and sensing (JCAS) has attracted extensive attention lately due to its potential of substantially improving the cost-, energy- and spectral-efficiency for a myriad of modern wireless systems that require both communications and radar, e.g., many smart IoT applications [1,2]. As a popular waveform in both communications and radar, the orthogonal frequency-division multiplexing (OFDM)-based JCAS has regained great interest after its arguable debut in 2007 [3]. The seminal work, however, did not illustrate OFDM radar sensing, and was only focused on the impact of radar antenna set-ups (unidirectional or omni-directional) on communication performances, e.g., bit error rate and system throughput. Before the work in [3], OFDM radar had been studied since 2000, yet without considering communications in general [4]. Early OFDM radar works between 2000 and 2009 mainly treat the OFDM waveform the same as the conventional radar waveforms, e.g., chirp, and intend to design OFDM-based waveforms, e.g., the phases of OFDM sub-carriers, to improve radar ambiguity functions [5,6,7,8,9,10]. Although many of these works [4,5,6,7,8,9,10] mention the applications of OFDM in data communications, they barely take into account any communication aspects, either in waveform design or in signal processing.

The true OFDM-based JCAS is enabled by the method first published in 2009 [11] and is more comprehensively elaborated on in [12]. At a radar receiver, the method [11] treats each OFDM symbol as in communication systems by first removing the cyclic prefix (CP) and then taking the discrete Fourier transform (DFT). In the frequency-domain, the method [11] removes the communication data symbols, as added on sub-carriers at the transmitter, through a point-wise division (PWD), attaining the scaled sum of the outer products between the range and Doppler steering vectors. A two-dimensional Fourier transform is then taken over the sub-carrier and time-domains, resulting in the so-called range–Doppler map (RDM), matrix or profile. Target detection and estimation can be performed using the RDM, which will be further illustrated in Section 3.

The sensing method [11,12] has been extensively applied in the past decade and has become a de facto standard for OFDM radar, particularly in automotive sensing [13,14,15,16,17,18,19,20,21,22,23]. For illustration convenience, we call the method [11] the classical OFDM sensing (COS) hereafter. Recent OFDM sensing works are mainly based on COS, but also introduce new techniques to improve the RDM quality. The work in [16] introduces the stepped carrier technique to increase the overall baseband bandwidth of the OFDM radar and, hence, the resolution of RDMs. The work in [17] randomizes the stepped carrier and exploits the compressive sensing technique to reconstruct a high-resolution RDM. Whereas previous works generally ignore the inter-carrier interference (ICI) issue, the work in [18] considers the impact of ICI on OFDM sensing and develops novel signaling, which repeats the same OFDM symbol over (slow-)time, to facilitate the estimation and suppression of ICI.

The OFDM sensing methods [11,12,16,17,18] reviewed above are for the single-antenna transceiver. One of the greatest advantages of using OFDM as a radar waveform is that multiple antennas can be utilized to realize orthogonal MIMO radar-like sensing. (In theory, given *M* transmitter antennas and *N* receiver antennas, an orthogonal multiple-input and multiple-output (MIMO) radar can achieve an extended virtual array of MN antennas.) This advantage is first noticed in [24], where an equidistant sub-carrier interleaving scheme is developed to make the signals transmitted by different antennas orthogonal. More specifically, the scheme makes antenna *m* only use sub-carriers m+iM for m=0,1,⋯,M−1 and i=0,1,⋯, where *M* is the antenna number. However, as noted in [19,20,21,22], the equidistant interleaving can reduce the unambiguously measurable distance of a MIMO-OFDM radar. To address the issue, a non-equidistant sub-carrier interleaving scheme is proposed in [19], where the genetic algorithm (GA) is used to maximize the ranging performance in terms of the sub-carrier interleaving patterns of transmitter antennas. In [20], the random time–frequency multiplexing is proposed to enhance the inter-antenna signal orthogonality of a MIMO-OFDM radar. For the same purpose, coded MIMO-OFDM radars are developed in [21,22], where special coding over time-, frequency-, space- and joint-domains are developed.

Targeted at sole radar applications, the methods reviewed above pay little attention to (MIMO-)OFDM data communications. Recently, the communication community has been highly active in promoting JCAS. In fact, given its potential in improving the cost-/energy-/spectral-efficiency and in substantially benefiting emerging use cases of mobile networks, e.g., smart home/city/ transportation [25], JCAS has been envisioned as a hallmark technology of the future sixth generation mobile communications (6G) [26]. The communication-centric JCAS designs in the communication community generally fall into two broad categories: general designs and communication standards-based ones.

The general JCAS waveforms have been designed in spatial, time and frequency-domains without referring to some specific communication standards. In the spatial-domain, dual-functional precoders/beamformers are generally designed to, e.g., approach desired sensing waveforms subject to signal-to-interference-plus-noise ratio (SINR) requirements for multi-user downlink MIMO communications [27]. In the time- and frequency-domains, existing works mainly resort to designing the frame structure [28], sub-carrier occupation [29], power allocation [30] and pilot/preamble signals [31]. These JCAS works [27,28,29,30,31] evaluate the sensing performance by statistical or asymptotic metrics, e.g., the signal-to-interference-plus-noise ratio (SINR) and the Cramer–Rao low bound (CRLB). They either do not discuss specific sensing methods or refer to some common ones, e.g., COS [11] reviewed above.

Standards-based JCAS designs generally integrate sensing into an existing communication system and prioritize communications. In this line of research, the IEEE 802.11ad-based millimeter-wave (mmWave) communication system is a popular choice. To counteract the severe attenuation of mmWave signals, IEEE 802.11ad mainly uses the DFT-spread OFDM (DFT-s-OFDM) waveform (DFT-s-OFDM is also known as the single-carrier OFDM (SC-OFDM). It performs DFT precoding before modulating data symbols onto sub-carriers and generally achieves a lower peak-to-average-power ratio (PAPR) than OFDM [32]) for data transmission. In [33,34], different sensing methods are developed using the Golay complementary sequences (GCSs) in the preamble of IEEE 802.11ad communication signals. In [35], the Doppler resilience of IEEE 802.11ad-based sensing is improved by incorporating Prouhet–Thue–Morse sequences in the preamble. In [31], an adaptive mmWave JCAS based on IEEE 802.11ad is developed, where a few non-uniformly placed preambles are transmitted to construct several receive virtual preambles for enhancing the velocity estimation accuracy at the cost of a small reduction in the communication data rate. While these methods exploit the superb auto-correlation feature of GCSs for a high ranging performance, it can be non-trivial to adapt them for other communication standards. This is more the case for existing wifi-based JCAS designs that mainly exploit the channel state information estimated by wifi devices [36].

The standards-based JCAS designs reviewed above exploit only a small portion of available signals in a standardized communication system. To further improve the sensing robustness against interference and noises, data signals of IEEE 802.11ad, with a much wider availability than preamble signals, are exploited for sensing in [37,38]. In [37], the generalized likelihood ratio test (GLRT) is employed to formulate a maximum likelihood (ML) problem for target detection and estimation. An adaptive algorithm is developed to solve the ML problem by iteratively estimating the current strongest target, reconstructing the target echo signal and removing it for estimating the next strongest target. While the method [37] results in a ML-like sensing performance, it has a much higher computational complexity than COS [11]. However, COS, if directly applied to DFT-s-OFDM, can cause a severe noise enhancement, as the communication signals modulated on sub-carriers approximately conform to a centered Gaussian distribution. To address the noise enhancement issue, the work in [38] modifies COS by replacing PWD with a point-wise product (PWP). Since the PWP of two frequency-domain signals plus a Fourier transform result in the cyclic cross-correlation (CCC) of the corresponding time-domain signals, we call the method [38] C-COS hereafter.

COS and C-COS have complexity only dominated by Fourier transforms. Thus, they particularly suit communication platforms needing (or benefiting from) radar sensing with limited computing ability, such as low-profile IoT devices. Although COS and C-COS have a sub-optimal sensing performance compared with the optimal ML estimation, they can provide a satisfactory sensing performance for numerous scenarios, such as detecting car presence in a car park or people presence indoors. Moreover, we can also perform COS and C-COS for initial sensing and can then exploit ML to refine the initial results. Such a combination can have much lower computational complexity than using ML directly. Further, as they do not make changes to communications, COS and C-COS allow for sensing to be added onto existing communication systems with minimal changes. Therefore, we envision that COS and C-COS will promisingly contribute to speeding up the market penetration of JCAS in the near future. This would be more the case if the following issues of COS and C-COS can be effectively addressed.
Passively reusing communication signals without making any changes makes COS and C-COS suffer from the sensing constraints imposed by communication signal formats. In particular, the maximum sensing distance is limited by the CP length of the underlying communication systems, and the maximum measurable velocity is inversely proportional to OFDM symbol duration. Thus, can we relieve the sensing limits without changing communication signal formats?COS and C-COS provide two different ways of generating RDMs. While their computational complexity is the same, a question follows naturally: which one gives the better sensing performance? It was shown through simulations in [38] that the C-COS can have a better sensing performance than COS in certain low SNR regions. This, however, is not always the case, as disclosed in our recent work [39]. A comprehensive analytical comparison between COS and C-COS is still missing;Can COS and C-COS be applied to future variants of OFDM? Recently, the orthogonal time–frequency space (OTFS) waveform has become increasingly popular due to its unique ability of handling fast time-varying channels. Like DFT-s-OFDM, OTFS is also a DFT-precoded OFDM. Unlike DFT-s-OFDM, which is only precoded once along the sub-carrier dimension, OTFS is additionally precoded over (slow-)time. However, directly applying COS or C-COS to OTFS can be hard, as the OTFS with a reduced cyclic prefix (RCP), i.e., a single CP for the whole block of OTFS symbols, is the main trend in the OTFS literature;Indeed, COS and C-COS already have quite a low computational complexity. However, should we rest on our laurels? In time-critical JCAS applications, we may require sensing to be carried out as fast as possible. This can be extremely challenging, particularly when the spatial volume to be sensed is large. All of these factors create highly stringent requirements for the sensing efficiency. Therefore, it is always beneficial to further reduce the sensing computational complexity, even only slightly.

We remark that the issues highlighted above have been rarely treated so far in the literature, including effective solutions. To raise awareness of these issues in the JCAS community, we will provide a short tutorial on them in Section 3 after we establish the signal model in Section 2. These two sections act as a fundamental basis to understand the recent progress and new solutions to be introduced sequentially. In particular, we will illustrate in Section 4 some recent research efforts, which are based on our own works [39,40], in addressing the first three issues mentioned above. Moreover, in Section 5, we will unveil that there exists non-trivial redundancy in OFDM-like sensing signals. To the best of our knowledge, such redundancy has not been explicitly treated in the literature yet. Noticing that, we develop a novel low-complexity sensing method based on COS by introducing efficient signal decimation. We also provide analysis and extensive simulations, demonstrating that the decimation-based COS can reduce the sensing complexity in a non-trivial manner, yet incurs only a minimal impact on the sensing performance.

## 2. Signal Model of OFDM-, DFT-s-OFDM- and OTFS-Based Sensing

Consider a general JCAS scenario where OFDM communication symbols are also used for sensing through a full-duplex synchronized receiver (Rx) co-located with the transmitter (Tx). We assume that proper full-duplex techniques are used to avoid/remove self-interference from Tx to Rx; see, e.g., [2] for a review of such techniques. In addition, single-antenna Tx and Rx are employed to introduce the core idea that is independent of spatial information in theory. Note that we start with OFDM for illustration clarity and will extend the signal model to DFT-s-OFDM and OTFS later.

For the *m*-th (m=0,1,⋯,M−1) OFDM symbol, there are *N* data symbols to be transmitted, as denoted by $s_{m}\left(n\right)\;\left(n=0,1,\cdots ,N-1\right)$. In OFDM, these *N* data symbols are multiplied onto *N* orthogonal sub-carriers, which essentially are single-tone signals at center frequencies of n/T. Here, *T* is the duration of the sub-carriers in the time-domain. This further indicates that the bandwidth of the considered OFDM system is B=N/T. Let $T_{s}$ denote the sampling time that satisfies $T_{s}=1/B=T/N$ in OFDM. Accordingly, the *m*-th OFDM symbol can be expressed as a discrete function of time index *k*, i.e.,
(1)$$\begin{array}{c} x_{m}\left(k\right)=\frac{1}{N}\sum_{n=0}^{N-1}s_{m}\left(n\right)e^{j2\pi nkT_{s}/T}=\frac{1}{N}\sum_{n=0}^{N-1}s_{m}\left(n\right)e^{j2\pi nk/N},\;k=0,1,\cdots ,N-1. \end{array}$$

From (1), we see that multiplying data symbols with *N* orthogonal sub-carriers is equivalent to taking the *N*-dimensional inverse DFT (IDFT) of the data symbols. In turn, taking the DFT of $x_{m}\left(k\right)$ with respect to (w.r.t.) *k* can recover $s_{m}\left(n\right)$.

According to the circular shift property [41], the DFT of any circularly shifted $x_{m}\left(k\right)$ is still $s_{m}\left(n\right)$, yet with extra phase shifts. Based on (1), we can write
(2)$$\begin{array}{c} x_{m}\left({\langle k-l\rangle }_{N}\right)=\frac{1}{N}\sum_{n=0}^{N-1}\left(s_{m}\left(n\right)e^{-j2\pi ln/N}\right)e^{j2\pi nk/N},\;k=0,1,\cdots ,N-1,\;\forall l \end{array}$$
where ${\langle \cdot \rangle }_{N}$ denotes modulo-*N*. Since the sample delay *l* resembles the echo delay in the sensing Rx, it is implied by (2) that *the sequence of $s_{m}\left(n\right)$ can always be recovered from the target echo as long as a complete (circularly shifted) OFDM symbol is available*. To ensure this, a CP is generally added to $x_{m}\left(k\right)$ by copying the last *Q* samples and pasting them to the beginning of $x_{m}\left(k\right)$; refer to Figure 1. Denoting the number of samples in the CP by *Q*, the *m*-th OFDM symbol becomes
(3)$$\begin{array}{c} {\tilde{x}}_{m}\left(\tilde{k}\right)=x_{m}\left({\langle \tilde{k}-Q\rangle }_{N}\right),\;\tilde{k}=0,1,\cdots ,N+Q-1, \end{array}$$
which is obtained by plugging $k={\langle \tilde{k}-Q\rangle }_{N}$ into (1). The timing relation between ${\tilde{x}}_{m}\left(\tilde{k}\right)$ and $x_{m}\left(k\right)$ is described in Figure 1.

Next, we build the signal model for target echoes. For illustration convenience and clarity, we model a single sensing target whose range, velocity and reflection coefficient are *r*, *v* and α, respectively. We also assume that *r*, *v* and α keep constant over *M* OFDM symbols, as complied with the Swerling-I target fluctuation model [42] [Ch.7]. The round trip (from Tx to the target and then back to Rx) causes a delay of $k_{r}=2r/(CT_{s})$ samples in the target echo, as compared with the transmitted OFDM symbol, where C is the microwave propagation speed. Note that $k_{r}$ may not be an integer. The target velocity incurs a Doppler frequency that can be calculated as $\mu =2vf_{c}/C$, where $f_{c}$ denotes the carrier frequency of the JCAS system. Taking into account $k_{r}$ and μ, the target echo can be modeled as
(4)$$\begin{array}{c} {\tilde{y}}_{m}\left(\tilde{k}\right)=\alpha g\left(\tilde{k}\right){\tilde{x}}_{m}\left(\tilde{k}-k_{r}\right)e^{j2\pi m\tilde{T}\mu },\;\tilde{k}=0,1,\cdots ,N+Q-1 \end{array}$$
where $g(\tilde{k})=0$ for $\tilde{k}=0,1,\cdots ,\left\lfloor k_{r}\right\rceil -1$ and $g(\tilde{k})=1$ for the remaining values of $\tilde{k}$; and $\tilde{T}=T+QT_{s}$ denotes the time duration of a CP-OFDM symbol. Here, ⌊x⌉ rounds *x* to the nearest integer. The echo timing with reference to the emitted signal is exemplified in Figure 1. Though noises are inevitable in any practical Rx, they are suppressed in (4) for brevity. Moreover, the “stop-and-hop” model [42] has been used to account for the Doppler effect by omitting the intra-symbol Doppler-related change. (Despite its wide applicability in conventional radar processing, the “stop-and-hop” model can be subjected to the condition that $\frac{4\pi v^{2}T_{\mathrm{CPI}}}{\lambda C}$ is less than a fraction of π radians, e.g., π/4 [43]. Here, $T_{\mathrm{CPI}}$ is a coherent processing time interval. Interested readers are referred to [43] [Chap. 2] for how the phase term is derived.)

*Extension to DFT-s-OFDM:* As mentioned in Section 1, DFT-s-OFDM is a DFT-precoded OFDM. The precoding happens along the sub-carrier-domain. Thus, instead of directly modulating communication data symbols onto sub-carriers, a DFT is taken first, and then the results are mapped to OFDM sub-carriers in interleaving or consecutive manners. Let ${\tilde{s}}_{m}\left(\tilde{n}\right)\;\left(\tilde{n}=0,1,\cdots ,\tilde{N}-1\right)$ be the communication data symbols to be transmitted, where $\tilde{N}$ is generally a fraction of *N*. Assume $\tilde{N}=\frac{N}{L}$, with *L* being an integer (related to the number of users in frequency division multiple access). Taking the $\tilde{N}$-point DFT of ${\tilde{s}}_{m}\left(\tilde{n}\right)$, we obtain ${\overset{˘}{s}}_{m}\left(\overset{˘}{n}\right)\;\left(\overset{˘}{n}=0,1,\cdots ,\tilde{N}-1\right)$. Then, we can map ${\overset{˘}{s}}_{m}\left(\overset{˘}{n}\right)$ onto *N* sub-carriers. In the interleaving mapping, we have
(5)$$\begin{array}{c} {\bar{s}}_{m}\left(n\right)={\overset{˘}{s}}_{m}\left(\overset{˘}{n}\right)\;\mathrm{for}\;n=l+\overset{˘}{n}L,\;\forall l=0,1,\cdots ,L-1. \end{array}$$

In the consecutive mapping, we have
(6)$$\begin{array}{c} {\bar{s}}_{m}\left(n\right)={\overset{˘}{s}}_{m}\left(\overset{˘}{n}\right)\;\mathrm{for}\;n=l\tilde{N}+\overset{˘}{n},\;\forall l=0,1,\cdots ,L-1. \end{array}$$

If multiple user-ends are served, they can be assigned with different *l*’s. Replacing $s_{m}\left(n\right)$ in (1) with ${\bar{s}}_{m}\left(n\right)$, the signal given in (4) also models the echo signal in DFT-s-OFDM sensing.

*Extension to OTFS:* Compared with DFT-s-OFDM, OTFS adds another DFT precoding over the slow time dimension. Let ${\bar{s}}_{m}\left(n\right)\;\left(m=0,1,\cdots ,M-1;\;n=0,1,\cdots ,N-1\right)$ denote the signal modulated onto sub-carriers. Different from OFDM, ${\bar{s}}_{m}\left(n\right)$ is not directly from a communication constellation and, instead, is now a two-dimensional symplectic Fourier transform of data symbols, as denoted by ${\tilde{s}}_{\tilde{m}}\left(\tilde{n}\right)\;\left(\tilde{m}=0,1,\cdots ,M-1;\;\tilde{n}=0,1,\cdots ,N-1\right)$. If no window function is used, ${\bar{s}}_{m}\left(n\right)$ is just the DFT of ${\tilde{s}}_{\tilde{m}}\left(\tilde{n}\right)$ over $\tilde{n}$ and the IFDT of the DFT results over $\tilde{m}$. Note that $\tilde{m}$ has a physical meaning of the Doppler index and $\tilde{n}$ the range index. They are dual variables of *m* (slow-time) and *n* (sub-carrier), respectively. Replacing $s_{m}\left(n\right)$ in (1) with ${\bar{s}}_{m}\left(n\right)$, we obtain the time-domain symbols. In CP-OTFS [44], each time-domain symbol is added with a CP, as shown in Figure 1. However, the OTFS with reduced CP (RCP-OTFS) is more popular in existing OTFS studies, as illustrated in Figure 2. Although the CP leads to a cyclically shifted version of the whole block of symbols at the sensing receiver, the ICI can be severe, particularly when the block duration is large. The severe ICI invalidates COS and C-COS for OTFS, as they implicitly require negligible ICI to generate RDMs. This will be made clear shortly in Section 3.

**Remark** **1.**
*For OFDM, the frequency-domain signals, i.e., those modulated onto sub-carriers, are independently drawn from a communication constellation, such as PSK and QAM. Thus, they conform to uniform distributions with a limited number of values. For DFT-s-OFDM and OTFS, however, their frequency-domain signals approximately conform to centered Gaussian distributions. This is because they are DFT(s) of the communication data symbols independently drawn from some constellations, while such DFT results converge in distribution to complex Gaussian random processes [45].*


## 3. COS and C-COS

In this section, we first review COS [11] and C-COS [38] based on the signal model established above. Then, we further illustrate the issues highlighted at the end of Section 1.

The diagram of the two methods is illustrated in Figure 3. They share the same signal preprocessing. Namely, they first remove the CP of each received symbol, i.e., ${\tilde{y}}_{m}\left(\tilde{k}\right)$ given in (4), and then transform the CP-removed symbol into the frequency-domain via a DFT.

From Figure 1, we see that the non-trivial part of ${\tilde{y}}_{m}\left(\tilde{k}\right)$ contains a circularly shifted OFDM symbol if $k_{r}\leq Q$ is satisfied, where $k_{r}$ is the target delay and *Q* is the CP length. Under the condition, removing the first *Q* samples of ${\tilde{y}}_{m}\left(\tilde{k}\right)$ yields ${\bar{y}}_{m}\left(k\right)=\alpha x_{m}\left({\langle k-k_{r}\rangle }_{N}\right)e^{j2\pi m\tilde{T}\mu }$ for k=0,1,⋯,N−1. By taking $l=k_{r}$ in (2), the DFT of $x_{m}\left({\langle k-k_{r}\rangle }_{N}\right)$ w.r.t. *k* is $s_{m}\left(n\right)e^{-j2\pi nk_{r}/N}$. Since $\alpha e^{j2\pi m\tilde{T}\mu }$ is a coefficient independent of *k*, the DFT of ${\bar{y}}_{m}\left(k\right)$ w.r.t. *k* can be directly given by ${\overset{˘}{y}}_{m}\left(n\right)=\alpha s_{m}\left(n\right)e^{-j2\pi nk_{r}/N}e^{j2\pi m\tilde{T}\mu }$. *The next step of removing the communication data symbol $s_{m}\left(n\right)$ differentiates COS and C-COS.*

**In COS**, PWD is used, leading to
(7)$$\begin{array}{c} y_{m}\left(n\right)={\overset{˘}{y}}_{m}\left(n\right)/s_{m}\left(n\right)=\alpha e^{-j2\pi nk_{r}/N}e^{j2\pi m\tilde{T}\mu }, \end{array}$$
where we assume that the sensing receiver has a copy of $s_{m}\left(n\right)$, as shown in Figure 3. Taking the 2D-DFT of $y_{m}\left(n\right)$ gives the following RDM,
(8)$$\begin{array}{c} Y_{b}\left(k\right)=\alpha \sum_{n=0}^{N-1}w_{N}\left(n\right)e^{-\frac{j2\pi nk_{r}}{N}}e^{-j\frac{2\pi kn}{N}}\times \sum_{m=0}^{M-1}w_{M}\left(m\right)e^{j2\pi m\tilde{T}\mu }e^{-j\frac{2\pi bm}{M}}, \end{array}$$
where $w_{N}\left(n\right)$ and $w_{M}\left(m\right)$ denote window functions of lengths *N* and *M*, respectively. If rectangular window functions are used, the *n*- and *m*-related summations will approach two sinc functions. They have mainlobes centered around $k=k^{†}={\langle N-\left\lfloor k_{r}\right\rceil \rangle }_{N}$ and $b={\tilde{b}}^{†}=\left\lfloor \mu \tilde{T}M\right\rceil$, and sidelobes elsewhere. As in the digital filter design, a proper window function, such chebshev, can be employed to suppress the sidelobes over *k* and *b* at the cost of an increased mainlobe width [41]. Given $k_{r}=\left\lfloor 2r/\left(CT_{s}\right)\right\rceil$ and μ=2v/λ, *r* and *v* can be estimated as
(9)$$\begin{array}{c} \hat{r}≊\left(N-k^{†}\right)CT_{s}/2,\;\hat{v}≊b^{†}C/\left(2Mf_{c}\tilde{T}\right),\;s.t.\;b^{†}=\left\{\begin{array}{cc} {\tilde{b}}^{†} & \mathrm{if}\;{\tilde{b}}^{†}\leq M/2\\ {\tilde{b}}^{†}-M & \mathrm{otherwise} \end{array}\right., \end{array}$$
where $b^{†}$ is a modified version of ${\tilde{b}}^{†}$ to account for negative velocities.

**In C-COS**, the PWD performed in (7) is replaced with PWP, which can be expressed as
(10)$$\begin{array}{c} z_{m}\left(n\right)={\overset{˘}{y}}_{m}\left(n\right)\times s_{m}^{*}\left(n\right)=\alpha {\left|s_{m}\left(n\right)\right|}^{2}e^{-j2\pi nk_{r}/N}e^{j2\pi m\tilde{T}\mu }, \end{array}$$
where ${\left(\right)}^{*}$ denotes a conjugate. Then, a 2D-DFT of $z_{m}\left(n\right)$ yields the following RDM,
(11)$$\begin{array}{c} Z_{b}\left(k\right)=\alpha \sum_{m=0}^{M-1}w_{M}\left(m\right)\left(\sum_{n=0}^{N-1}w_{N}\left(n\right){\left|s_{m}\left(n\right)\right|}^{2}e^{-\frac{j2\pi nk_{r}}{N}}e^{-j\frac{2\pi kn}{N}}\right)e^{j2\pi m\tilde{T}\mu }e^{-j\frac{2\pi bm}{M}}, \end{array}$$
where $Z_{b}\left(k\right)\neq Y_{b}\left(k\right)$ if $|s_{m}\left(n\right)|\neq 1$; otherwise $Z_{b}\left(k\right)=Y_{b}\left(k\right)$. Note that, if $w_{N}\left(n\right)=1$, the *n*-related summation in $Z_{b}\left(k\right)$ can be rewritten into
(12)$$\begin{array}{cc} \; & \sum_{n=0}^{N-1}s_{m}\left(n\right)s_{m}^{*}\left(n\right)e^{-\frac{j2\pi nk_{r}}{N}}e^{-j\frac{2\pi kn}{N}}\overset{\left(a\right)}{=}\sum_{n=0}^{N-1}\sum_{k^{\prime }=0}^{N-1}{\tilde{s}}_{m}\left({\langle k^{\prime }-k_{r}\rangle }_{N}\right)e^{-j\frac{2\pi k^{\prime }n}{N}}s_{m}^{*}\left(n\right)e^{-j\frac{2\pi kn}{N}}\\ \; & \overset{\left(b\right)}{=}\sum_{k^{\prime }=0}^{N-1}{\tilde{s}}_{m}\left({\langle k^{\prime }-k_{r}\rangle }_{N}\right){\tilde{s}}_{m}^{*}\left({\langle k^{\prime }+k\rangle }_{N}\right), \end{array}$$
where ${\tilde{s}}_{m}\left(k\right)$ denotes the IDFT of $s_{m}\left(n\right)$ (which is the frequency-domain signal), $\overset{\left(a\right)}{=}$ is obtained by replacing $s_{m}\left(n\right)$ with its DFT expression, i.e., the $k^{\prime }$-summation in the middle result, and $\overset{\left(b\right)}{=}$ is because the *n*-summation can be seen as the conjugate of the IDFT of $s_{m}\left(n\right)$. Note that the last result is the CCC of ${\tilde{s}}_{m}\left(k\right)$ and ${\tilde{s}}_{m}\left({\langle k-k_{r}\rangle }_{N}\right)$. Thus, the *n*-related summation in (11) resembles the matched filtering in the conventional radar signal processing.

As illustrated in Remark 1, $s_{m}\left(n\right)$ approximately conforms to a centered Gaussian distribution. Since IDFT is a unitary transformation, ${\tilde{s}}_{m}\left(k\right)$, as the IDFT of $s_{m}\left(n\right)$, is also a centered Gaussian signal. Thus, the CCC result given in (12) will present a mainlobe around $k=k^{†}={\langle N-\left\lfloor k_{r}\right\rceil \rangle }_{N}$, which is the same as in COS. The difference is that we do not have an analytical model to depict the CCC result. Moreover, the sidelobe levels in the CCC result are unpredictable; c.f., the deterministic sidelobes of $|Y_{b}\left(k\right)|$ over *k*’s. On the other hand, comparing (8) and (11), the two RDMs share the same Doppler measurement ability, which is solely dependent on the *m*-related summation. Based on the above elaboration, we conclude that the estimates given in (9) also apply to C-COS.

Figure 4a illustrates the RDM $Y_{m}\left(b\right)$ by plotting its amplitude against range and Doppler grids. Three targets are set, with parameters summarized in Table 1. From Figure 4a, we see three mainlobes corresponding to three targets. Based on the illustration below (8), the indexes of the range grids of the three targets can be calculated as ${\langle N-\left\lfloor k_{r}\right\rceil \rangle }_{N}\times 16=4064$, 4040 and 4016, where multiplying 16 is due to the increasing of the DFT size (as illustrated in the caption of Figure 4). Similarly, we can calculate the indexes of the Doppler grids of the targets, as given by $\lfloor \mu \tilde{T}M\rceil \times 16=32,32$ and 64. We see from Figure 4a that the peak locations match the above calculations.

**Target detection:** If there exists a single target, detecting the target can be readily achieved through identifying the peak of $|Y_{b}\left(k\right)|$ or $|Z_{b}\left(k\right)|$. The single-target scenario may sound unrealistic to the conventional radar community. However, in mmWave JCAS, the single-target sensing can be practical and has been studied in [31,33,37]. To counteract the severe path loss of mmWave signals, mmWave communication systems generally use large-scale antenna arrays to steer highly directional beams. Therefore, a mmWave signal in the beam direction is likely to be blocked by the first target.

In multi-target scenarios, there can be multiple mainlobes in the RDM. Directly identifying the peak of the RDM will detect the strongest target. Then, the parameters of the strongest target can be estimated based on (9). With its parameter estimates, the target can be reconstructed and removed from the RDM, enabling the detection of the next strongest target. Such a sequential detection can be time-consuming, as the detection of each target will involve searching over the whole range-Doppler space.

Classical radar detectors can be employed to detect multiple targets efficiently. The constant false-alarm rate (CFAR) detector is one of the most commonly used radar detectors [42]. Briefly speaking, CFAR tests all range and Doppler grids, as indexed by *k* and *b*, respectively, to check the presence of a target. At a grid under test (GUT), CFAR calculates the background interference-plus-noise (IN) power by averaging the power of the grids around GUT. The adjacent grids around GUT are generally excluded from the power evaluation to reduce the impact of the strongest sidelobes of a target. The estimated IN is amplified by a coefficient and used as a threshold, where the coefficient is dependent on the expected false-alarm rate. If the power of the GUT is greater than the threshold, then CFAR reports the presence of a target at the GUT; otherwise, CFAR reports target absence. A simulation tutorial of CFAR is provided by MathWorks in [46].

**Target estimation:** After targets are detected, their locations can be submitted to (9) for estimating their parameters. However, the range and velocity estimations obtained in (9) suffer from errors as large as $CT_{s}/4$ and $C/(4Mf_{c}\tilde{T})$, respectively. In fact, we can further refine the estimations using, e.g., the classical multiple signal clarification (MUSIC) algorithm [47]. Some newer estimators [48,49,50], which are efficient with low complexity, are also good candidates for refining target estimations. These estimators interpolate DFT coefficients around the integer range–Doppler grid of a target, as obtained in the target detection, and then solve the accurate parameter estimations from a pre-established relation between the interpolations and target parameters. Interested readers are referred to [48,49,50] for more details.

**CP-limited maximum measurable range (MMR):** To obtain the RDMs in (8) and (11), we have assumed that the maximum target delay is no greater than the CP duration, *Q*. This, in turn, indicates that the MMR of COS and C-COS is limited by CP. In particular, it can be given by $CQT_{s}/2$, where *Q* is the CP length. Unfortunately, such a limitation remains for COS and C-COS, even when we have a sufficient link budget for sensing a longer distance. In essence, the MMR limitation exists because each communication symbol, e.g., in Figure 1 and Figure 2, is treated as an independent sensing waveform, and the zero inter-symbol interference is pursued in COS and C-COS. If we treat a block of consecutive symbols, e.g., symbol 0,⋯,M−1 in Figure 2, as a single waveform, and use the whole signal block as a matched filter coefficient to process the received echo signal, the MMR limitation discussed here may be lifted. This whole-block processing, however, can suffer from a non-negligible intra-block Doppler impact. A pointwise Doppler compensation can be performed before range processing [14,18]. Moreover, a two-dimensional maximum likelihood-based range and Doppler simultaneous estimation can also be performed [34,51]. These options generally have a non-trivially higher complexity than COS and C-COS.

**Symbol-limited maximum measurable Doppler (MMD):** From the echo signal model given in (4), we see that the sampling interval over the slow time is $\tilde{T}=\left(N+Q\right)T_{s}$, which is dominated by the OFDM symbol duration. The sampling frequency is $1/\tilde{T}$. Then, the maximum measurable Doppler frequency is given by $1/(2\left(N+Q\right)T_{s})$. Moreover, the Doppler resolution can be given by $1/(M\left(N+Q\right)T_{s})$, which is inversely proportional to the overall time of a whole block.

**COS versus C-COS:** As shown in Figure 3, COS and C-COS are differentiated by the way they handle frequency-domain communication signals. Recall that COS and C-COS apply PWD and PWP, respectively, as given in (8) and (11). When the communication signals have a constant modulus in the frequency-domain, PWD and PWP yield the same result. However, for DFT-s-OFDM and OTFS, the communication signals conform to a centered Gaussian distribution, as illustrated in Remark 1. In such cases, PWD can severely enhance the background noise in the RDM, as the signal being or approaching zero is used as a divisor; see (8) (For illustration simplicity, we ignored the noise term in (8), though it is inevitable in practice). PWP is proposed to relieve noise enhancement [38]. Comparing Figure 4b,c, we see that PWP indeed leads to a smaller noise background in the RDM. However, PWP can lead to a non-negligible noise floor in moderate and high SNR cases. Thus, analytical comparisons between PWP and PWD are worth investigating to provide the guidance in an ad hoc selection between them.

## 4. Recent Progress

The three issues discussed at the end of the last section have rarely been noticed in the literature, not to mention any solutions. Recently, we have performed some preliminary studies on relieving the issues [39,40]. In this section, we highlight some interesting results and remaining challenges.

Whether the waveform is OFDM, DFT-s-OFDM or OTFS, we are actually facing the same problem: *to detect and estimate targets given a block of communication signals, e.g., the middle signals in Figure 1 and Figure 2, and the target echo signals, e.g., the lower signals in Figure 1 and Figure 2.* We emphasize that, in the considered JCAS, we do not intend to make any changes to the underlying communication system. In the COS and C-COS reviewed earlier, they segment the communication signals at a sensing receiver by fully complying to communications format, i.e., (N+Q) samples a symbol and *M* symbols in total. From the end of Section 3, we have seen that such compliance is the root of the sensing restrictions.

In light of the above observation, we proposed a novel sensing framework recently in [39]. Here, we unitedly use x(i) to denote the communication-transmitted signal in the time-domain, where the communication system can be based on either OFDM, DFT-s-OFDM or OTFS. That is, x(i) can be the middle signal in either Figure 1 or Figure 2. Moreover, we point out that x(i) is a signal sequence with i=0,1,⋯,I−1. In the case of CP-OFDM and DFT-s-OFDM, we have I=M(N+Q)−1, where *M* is the number of symbols and (N+Q) is the number of samples per symbol (including CP); see Figure 1. In the case of RCP-OTFS (as illustrated at the end of Section 2), we have I=MN+Q−1, where a single CP of *Q* samples is applied to a block of *M* symbols; see Figure 2. As shown on the left of Figure 5, x(i) hits targets and propagates to the sensing receiver, resulting in the echo signal denoted by y(i). Thus, y(i) is the scaling of the time-delayed x(i), similar to the relation between ${\tilde{y}}_{m}\left(\tilde{k}\right)$ and ${\tilde{x}}_{m}\left(\tilde{k}-k_{r}\right)$ depicted in (4). As mentioned in Section 2, the co-located transceiver is considered in this paper. Thus, it is reasonable to assume that the sensing receiver shares the same clock as the transmitter and has a copy of x(i) stored for sensing processing.

The sensing framework [39] starts with a segmentation, as performed on both x(i) and y(i) in the same way. In particular, we ignore the signal format of the underlying communication system and segment x(i) and y(i) in a sensing-favorable manner. As shown in Figure 5, x(i) and y(i) are segmented into consecutive sub-blocks (SBs) evenly, with $\tilde{N}$ samples per segment, where $\tilde{N}=N$ is no longer necessary. Adjacent SBs are allowed to overlap for $\bar{Q}\left(\geq 0\right)$ samples. Let $x_{m}\left(n\right)$ and $y_{m}\left(n\right)$ denote the signal in the *m*-th SB. Due to target delay, $y_{m}\left(n\right)$ only contains a part of $x_{m}\left(n\right)$, with the remaining part right after $y_{m}\left(n\right)$; see the illustration in Figure 5. Thus, we can add the $\tilde{Q}$ samples right after $y_{m}\left(n\right)$ onto the beginning of $y_{m}\left(n\right)$, making $y_{m}\left(n\right)$ contain cyclically shifted versions of $x_{m}\left(n\right)$. This will require $\tilde{Q}$ to be no smaller than the maximum target delay. Clearly, the $\tilde{Q}$ samples have a similar role as the CP in OFDM. Thus, we call them the virtual CP (VCP). However, we emphasize that VCP is not related to the original CP in any way. A key difference between them is that CP is determined by the communication system, but VCP is designated at the sensing receiver for sensing purposes. However, as shown in Figure 5, adding VCP can introduce inter-SB interference, which is the price paid for pursuing flexible sensing.

The DFT results in Figure 5 can be input to PWD and PWP for generating RDMs. For clarity, we summarize the novel sensing framework in Algorithm 1. In Step 1, the *m*-th SB starts at the $m(\tilde{N}-\bar{Q})$-th sample and has $\tilde{N}$ samples. Given *I* signal samples in x(i) and based on the illustration in the right sub-figure of Figure 5, the number of SBs is $\tilde{M}=\left\lfloor \frac{I-\tilde{Q}-\bar{Q}}{\tilde{N}-\bar{Q}}\right\rfloor$, where ⌊·⌋ takes flooring. In Step 2, VCP is added for the echo signal so that the *m*-th SB of the echo signal becomes underlain by the *m*-th SB of the transmitted signal, as illustrated in the right sub-figure of Figure 5. Steps 3–5 are the same as the last three steps of COS and C-COS illustrated in Figure 3. However, by introducing the novel segmentation and VCP, an ad hoc adjustment can be made to the sensing framework, and, hence, there is better catering for different sensing scenarios. For example, we can increase $\tilde{Q}$ for sensing a longer distance, we can reduce $\tilde{N}$ to increase the maximum measurable Doppler frequency and we can adjust $\bar{Q}$ in accordance with $\tilde{N}$ and $\tilde{Q}$ to improve sensing SINR. Next, we provide more elaborations on how to determine these key parameters.

**Remark** **2.**
*The criteria of configuring the sensing framework are illustrated below. First, we can set $\tilde{Q}$ based on $r_{\max}$, the required MMR. As it is related to the VCP length $\tilde{Q}$, the MMR can be given by $\frac{C\tilde{Q}T_{s}}{2}$, which, equating with $r_{\max}$, yields $\tilde{Q}=2r_{\max}/CT_{s}$. Second, we determine $\tilde{M}$ given the requirements on the velocity measurement. The maximum measurable Doppler frequency is half the sampling frequency over SBs (equivalent to the slow time in radar processing), which is $1/(2\left(\tilde{N}-\bar{Q}\right)T_{s})$. Here, $\left(\tilde{N}-\bar{Q}\right)T_{s}$ is the difference between the starting times of any two adjacent SBs (and hence the sampling time over the slow-time-domain); see Figure 5. Consequently, to cater for the maximum measurable Doppler, as denoted by $\mu_{\max}$, we need to keep $\tilde{N}\leq 1/\left(2\mu_{\max}T_{s}\right)+\bar{Q}$. Third, given $\tilde{N}$, we can then set $\bar{Q}$. To increase the SINR in both RDMs, we expect to have $\bar{Q}$ as large as possible; see (13) and (14), which are to be illustrated shortly. However, the larger the $\bar{Q}$, the more correlated the signals between adjacent sub-blocks can be; see Figure 5. The correlation can make the results in (13) and (14) less precise. The detailed impact, however, is difficult to analyze. As shown through the simulations in [39], the SINRs in (13) and (14) are consistently precise, even when $\bar{Q}$ takes as large as $\tilde{M}/2-\tilde{Q}$.*



**Algorithm 1** A novel sensing framework [39].*Input:*x(i)(i=0,1,⋯,I−1) (a copy of communication-transmitted signal sequence), y(i) (echo signal at a sensing receiver), $\tilde{N}$ (SB length), $\tilde{Q}$ (VCP length), $\bar{Q}$ (overlapping between adjacent SBs)
Segment x(i) and y(i) evenly into consecutive sub-blocks (SBs), as given by $x_{m}\left(n\right)$ and $y_{m}\left(n\right)$. The *m*-th $\left(m=0,1,\cdots ,\tilde{M}\right)$ sub-block consists of samples $i=m\left(\tilde{N}-\bar{Q}\right)+\left(0,1,\cdots ,\tilde{N}-1\right)$;Add VCP, i.e, the $\tilde{Q}$ samples right after $y_{m}\left(n\right)$, onto the beginning of $y_{m}\left(n\right)$, leading to ${\tilde{y}}_{m}\left(n\right)$;Take the DFT (w.r.t. *n*) of $x_{m}\left(n\right)$ and ${\tilde{y}}_{m}\left(n\right)$, yielding $X_{m}\left(k\right)$ and $Y_{m}\left(k\right)$;Under PWD, we have ${\tilde{U}}_{m}\left(k\right)=Y_{m}\left(k\right)./X_{m}\left(k\right)$, whereas using PWP, we obtain ${\tilde{V}}_{m}\left(k\right)=Y_{m}\left(k\right)\times X_{m}^{*}\left(k\right)$;Taking the 2D-DFT of ${\tilde{U}}_{m}\left(k\right)$ and ${\tilde{V}}_{m}\left(k\right)$, generate the RDMs $U_{b}\left(n\right)$ and $V_{b}\left(n\right)$, respectively.



The analytical SINRs of the two RDMs are helpful in investigating the sensing framework. The SINR of $U_{b}\left(n\right)$, as obtained in Step 5 of Algorithm 1, can be expressed as [39] [(33)]
(13)$$\gamma_{U}\left\{\begin{array}{c} \overset{\gamma_{0}\ll \frac{1}{\sigma_{P}^{2}}}{\approx }\frac{\tilde{N}\left(\frac{\left(I-\tilde{Q}-\bar{Q}\right)}{\left(\tilde{N}-\bar{Q}\right)}\right)\gamma_{0}\sigma_{P}^{2}}{\left(1+\frac{\tilde{Q}}{\tilde{N}}\right)b\left(\epsilon \right)}\overset{\left(a\right)}{\approx }\frac{I\gamma_{0}\sigma_{P}^{2}}{\left(1-\frac{\bar{Q}}{\tilde{N}}\right)\left(1+\frac{\tilde{Q}}{\tilde{N}}\right)b\left(\epsilon \right)}\\ \overset{\gamma_{0}\gg \frac{1}{\sigma_{P}^{2}}}{\approx }I/\left(\left(1-\frac{\bar{Q}}{\tilde{N}}\right)\frac{\tilde{Q}}{\tilde{N}}b\left(\epsilon \right)\right) \end{array}\right.,\;s.t.\;b\left(\epsilon \right)=2\ln\left(\frac{2(1-\epsilon )}{\left(e\sqrt{\epsilon (2-\epsilon )}\right)}\right),$$
where $\gamma_{0}$ is the SNR in the time-domain echo signal, i.e., y(i) in Algorithm 1; *I* is the total number of samples of y(i); $\sigma_{P}^{2}$ is the total power of targets; e is the natural number; and ϵ is a sufficiently small number, e.g., 1/I. Note that b(ϵ) accounts for the noise enhancement when diving a centered Gaussian signal by another one; see [40] for a detailed analysis of this issue. The SINR of $V_{b}\left(n\right)$, as obtained in Step 5 of Algorithm 1, can be expressed as [39] [(35)]
(14)$$\gamma_{V}\left\{\begin{array}{c} \overset{\gamma_{0}\ll \frac{1}{\sigma_{P}^{2}}}{\approx }I\gamma_{0}\sigma_{P}^{2}/\left(\left(1-\frac{\bar{Q}}{\tilde{M}}\right)\left(1+\frac{\tilde{Q}}{\tilde{M}}\right)\right)\\ \overset{\gamma_{0}\gg \frac{1}{\sigma_{P}^{2}}}{\approx }I/\left(\left(1-\frac{\bar{Q}}{\tilde{M}}\right)\left(1+\frac{\tilde{Q}}{\tilde{M}}\right)\right) \end{array}\right..$$

**Remark** **3.**
*Based on (13) and (14), we can make the following comparisons between the PWD- and PWP-based RDMs:*
*(3a)* 
*In low SNR regions where $\gamma_{0}\ll 1/\sigma_{P}^{2}$, $V_{b}\left(n\right)$ has an SINR that is b(ϵ) times the SINR in $U_{b}\left(n\right)$, where b(ϵ)>1 in general;*
*(3b)* 
*In high SNR regions where $\gamma_{0}\gg 1/\sigma_{P}^{2}$, $U_{b}\left(n\right)$ can have a greater SINR than $V_{b}\left(n\right)$, provided $b\left(\epsilon \right)\leq \frac{\tilde{M}}{\tilde{Q}}$;*
*(3c)* 
*Regardless of $\gamma_{0}$, the $V_{b}\left(n\right)$ always has a greater SINR than $U_{b}\left(n\right)$, if $b\left(\epsilon \right)>\frac{\tilde{M}}{\tilde{Q}}+1$.*


*The comparisons made above are helpful in selecting between PWD and PWP when generating RDMs.*


Before ending the section, we use a set of simulation results to showcase the superiority of the sensing framework illustrated in Algorithm 1 over conventional OFDM sensing (COS) in terms of the maximum measurable range. The simulation parameters are given in Table 2. Note that the CP length *Q* is much smaller than the sample delay of the target, i.e., $k_{r}$. This setting is particularly employed to validate the point made at the end of Section 3 (in terms of the limited maximum measurable range of COS and C-COS). Based on the review on COS and C-COS given in Section 3, we know that these conventional methods would not be able to sense the target set in Table 2. In contrast, the sensing framework given in Algorithm 1 can flexibly set $\tilde{Q}$ and $\tilde{N}$ according to Remark 2 so as to cater to different sensing needs. In particular, to sense $k_{r}=320$, we set $\tilde{Q}=321$, $\tilde{N}=2\tilde{Q}=642$ and $\bar{Q}=128$. The settings further lead to $\tilde{M}=78$. For convenience, random signals, conforming to a centered Gaussian distribution with the unit variance, are loaded onto the OFDM sub-carriers. This essentially simulates DFT-s-OFDM, as the frequency-domain signal presents such randomness according to Remark 1. Thus, C-COS is used to simulate the conventional OFDM sensing.

Figure 6 compares the RDMs generated by C-COS and the novel sensing framework (NSF). The results in the first row are obtained by C-COS. We see from Figure 6a that the RDM of C-COS is noise-like over the whole range–Doppler bins. Then, a close look at the range–Doppler bins around the theoretical target location is provided in Figure 6b. We still see no obvious target. The theoretical range bin of the target is $k_{r}=320$. Since it is greater than *N*, a modulo is taken, leading to 64. Based on the elaboration right after (8), the theoretical Doppler bin of the target can be calculated as $\lfloor \mu M\left(N+Q\right)/f_{s}\rceil =5$. The range and Doppler cuts of the RDM of C-COS are given in Figure 6c,d, respectively. Again, we do not see any obvious targets. That is, C-COS fails to detect the target set in Table 2.

The results in the second row are obtained by the NSF. Substantially different from Figure 6a,e presents a normal RDM with the target shown as a sharp peak. The theoretical range bin here is the same as that for C-COS, i.e., 320. The theoretical Doppler bin of NSF needs to be recalculated as $\lfloor \mu \tilde{M}\left(\tilde{N}-\bar{Q}\right)/f_{s}\rceil =5$ (the result, however, is the same). Figure 6f zooms in on the RDM around the bin pair (321,6) (both theoretical bins are added by one due to the fact that the MATLAB index starts from one and not zero in our calculation). From Figure 6f, we can see that the target peak is approximately 40 dB stronger than the background noise. This strongly contrasts with Figure 6b, validating the significant improvement of the novel sensing framework over the conventional COS. Range and Doppler cuts of NSF are given in Figure 6g,h, respectively. We clearly see strong peaks at target locations.

There is a remaining issue of NSF on false targets. We see from Figure 6g that, other than the true target at the 321-th range bin, there are two other weaker targets, which are located at the 65-th and 577-th range bins. These numbers have implicit relations with 321. Specifically, we have 321+N(=256)=577 and 312−N=65. These fake targets are generated due to the partial periodicity shown in the signal after adding VCPs; see Step 2 in Algorithm 1. The issue was also revealed in [39]. However, to date, there is no solution yet. One potential solution of suppressing fake targets is to employ the special relation between the locations of fake targets and that of the true target, in combination with the amplitude and phase features of their peaks. Another potential way of suppressing fake targets is to design NSF parameters, $\tilde{N}$, $\tilde{Q}$ and $\bar{Q}$, so that the partial periodicity leading to the fake targets can be removed. Validating these solutions or others calls for more research efforts.

## 5. A Novel Design to Further Reduce Sensing Complexity

We proceed to introduce an efficient design that further reduces the sensing complexity. Let us revisit the PWD-processed echo signal in COS, i.e., $y_{m}\left(n\right)$, as originally given in (7) and rewritten below
(15)$$\begin{array}{c} y_{m}\left(n\right)=\alpha e^{j2\pi m\tilde{T}\mu }e^{-j2\pi \left(nT_{s}\right)\frac{k_{r}}{NT_{s}}}=\alpha e^{j2\pi m\tilde{T}\mu }e^{-j2\pi \left(nT_{s}\right)\frac{k_{r}B}{N}}, \end{array}$$
where the last result is due to $B=1/T_{s}$. From the above expression, we see that the frequency of $y_{m}\left(n\right)$ is $k_{r}B/N$. As underlined in Section 3, COS requires $k_{r}\leq Q$. This indicates that the bandwidth of $y_{m}\left(n\right)$ is no greater than QB/N=B/D, where D=N/Q. In OFDM communication systems, Q≪N is generally satisfied [32]. Thus, we make the following assertion: *Provided that the maximum sample delay in the target echo is no greater than the CP length, and that the CP length is much less than the sub-carrier number, the PWD-processed echo signal has a much smaller bandwidth than an OFDM symbol.* Although we base our illustration on COS, the analysis and method are also applicable to C-COS and the novel sensing framework given in Algorithm 1.

Based on the signal models in Section 3, the above assertion can be interpreted as: provided $k_{r}\leq Q\ll N$, $y_{m}\left(n\right)$ has a much smaller bandwidth than $x_{m}\left(k\right)$ given in (1). With this fact noted, we can further conclude that only 1/D of the whole frequency band contains useful information for sensing, and that the rest is filled with noises. In other words, $y_{m}\left(n\right)$ can have considerably redundant information. To this end, *we propose decimating $y_{m}\left(n\right)$ to remove the inherent redundancy and, hence, to reduce the number of signal samples along the n-dimension, prior to sensing.* The decimation leads to a smaller RDM and, hence, reduces the complexity of RDM-dependent target detection and estimation.

**Remark** **4.**
*The assertion made for $y_{m}\left(n\right)$ also applies to $z_{m}\left(n\right)$, the PWP-processed signal given in (10), and ${\tilde{y}}_{m}\left(n\right)$, the VCP-added signal obtained in Step 2 of the algorithm summarized in Algorithm 1. Therefore, the decimation proposed above can also benefit C-COS and the novel sensing framework in Algorithm 1 in reducing the RDM dimension and the complexity of target detection/estimation.*


### 5.1. Efficient Decimation

We proceed to illustrate the efficient implementation of the proposed decimation. As seen from (15), $y_{m}\left(n\right)$ is a bandpass signal with frequency band [−B/D,0]. To decimate $y_{m}\left(n\right)$ by the factor of *D*, we develop the following procedure, as illustrated in Figure 7a.

(1)Anti-aliasing filtering: is performed on $y_{m}\left(n\right)$ to suppress out-of-band interference and noises. The passband of the filter is the same as that of $y_{m}\left(n\right)$, whereas the stopband is given by [−B/2,B/2]⊘[−B/D,0], where ⊘ denotes a set difference. The frequency spectrum of an ideal bandpass filter is shown in Node B of Figure 7b. The signal spectrum before and after filtering is shown in Nodes A and C of Figure 7b, respectively. As ideally illustrated in Node C, out-of-band noises are totally removed, which is impractical, but can be well approximated by designing the anti-aliasing filter with a large stopband attenuation;(2)Downsampling: is denoted by “D↓” in Figure 7a. It keeps every *D*-th sample (starting from sample 0) and deserts others. After downsampling, the sampling frequency is reduced to B/D, and the spectrum center becomes −B/(2D); see Node D of Figure 7b;(3)Frequency shifting: shifts the spectrum center of the downsampled signal to zero, which leads to the spectrum shown in Node E of Figure 7b.

Above are the general steps of a bandpass decimation. By invoking the polyphase structure, the decimation can be implemented more efficiently.

At the core of the polyphase structure is the decomposition of the anti-aliasing filter. Consider an (L−1)-order finite impulse response anti-aliasing filter. Let h(l) denote the *l*-th (l=0,1,⋯,L−1) filter coefficient. The *z*-transform of h(l) can be expressed as [52] [Ch.6]
(16)$$\begin{array}{c} H\left(z\right)=\sum_{l=0}^{L-1}h\left(l\right)z^{-l}=\sum_{d=0}^{D-1}z^{-d}\sum_{p=0}^{P-1}h\left(d+pD\right)z^{-pD}=\sum_{d=0}^{D-1}z^{-d}H_{d}\left(z^{D}\right), \end{array}$$
where the second equality is obtained by decomposing l=d+pD and the *p*-related summation is denoted by $H_{d}\left(z^{D}\right)$ in the last result. Note that L=PD is assumed in the above decomposition. The condition can be readily satisfied by specifying the filter order as (PD−1) when designing the anti-aliasing filter. Based on (16), we can see that the filter can be implemented in *D* parallel branches, as illustrated in Figure 7c. The input signal $y_{m}\left(n\right)$ goes into different branches simultaneously, and the outputs of the branch-filters, denoted by $H_{d}\left(z^{D}\right)\;\left(\forall d\right)$, are supposed to be summed and then downsampled. However, in Figure 7c, we move the downsampler to before the summation and equivalently put a downsampler in each branch. Carrying this out allows us to invoke the notable identity, as illustrated in Figure 7c, to exchange the orders of the filter and downsampler in each branch. The order exchanging makes the delay block, $z^{-d}$, adjacent to a downsampler. To this end, the samples to be filtered by the *d*-th (∀d) branch-filter become $y_{m}\left(\tilde{D}-d+qD\right)\;\left(q=0,1,\cdots ,Q-1\right)$, where “−d” reflects the *d*-delay block in branch *d*, $\tilde{D}=\left(D-1\right)$ is added to sample indexes to ensure that the indexes are no less than zero and qD is a result of the downsampler. Based on (16), the coefficients of the *d*-th branch-filter are $h_{d+pD}\;\left(p=0,1,\cdots ,P-1\right)$.

The filter decomposition and the order exchanging illustrated above lead to the polyphase structure of bandpass decimation, as shown in Figure 7d. In the figure, we use a buffer to collect continuous *Q* samples, i.e., $y_{m}\left(\tilde{D}-d+qD\right)\;\left(q=0,1,\cdots ,Q-1\right)$ for the *d*-th branch, and each branch-filter is implemented in the frequency-domain due to the following relation
$$\begin{array}{c} h_{d+pD}⊛y_{m}\left(\tilde{D}-d+qD\right)\equiv {\mathrm{IFFT}}_{\tilde{Q}}\left\{{\mathrm{FFT}}_{\tilde{Q}}\left\{h_{d+pD}\right\}⊙{\mathrm{FFT}}_{\tilde{Q}}\left\{y_{m}\left(\tilde{D}-d+qD\right)\right\}\right\}, \end{array}$$
where “⊛” denotes linear convolution, “≡” means that the calculations on its two sides are equivalent, ${\mathrm{IFFT}}_{\tilde{Q}}$ and ${\mathrm{FFT}}_{\tilde{Q}}$ denote size-$\tilde{Q}$ IFFT and FFT, respectively, and “⊙” calculates the point-wise product. Note that the above equivalence requires $\tilde{Q}\geq \left(P+Q-1\right)$. For radix-2 (I)FFT, we can take $\tilde{Q}$, such that $\log_{2}\tilde{Q}=\left\lceil \log_{2}\left(P+Q-1\right)\right\rceil$. Since each branch-filter produces (P−1) transient outputs and takes *Q* samples as an input, the indexes of valid filter outputs are P−1,P,⋯,Q−1. Thus, we keep the valid outputs and dump others, as shown in Figure 7d).

Referring back to Figure 7a, we are now at the last step of decimation, i.e, shifting the filtered and downsampled signal to the baseband. To differentiate with $y_{m}\left(n\right)$, we use $\hat{n}$ to denote the index of valid samples after downsampling, as also highlighted in Figure 7d. Based on (15), the filtered signal, after removing transients, can be expressed as
$$\begin{array}{c} \alpha e^{j2\pi m\tilde{T}\mu }e^{-j2\pi \hat{n}Dk_{r}/N}=\alpha e^{j2\pi m\tilde{T}\mu }e^{-j2\pi \hat{n}k_{r}/Q},\;\hat{n}=0,1,\cdots ,Q-P. \end{array}$$

As a discrete function of $\hat{n}$, the spectrum center of the above signal is now at π, since the mean value of $k_{r}$ is Q/2. According to the frequency shift property of Fourier transform, we know that an angular frequency shift of π can be equivalently realized by multiplying the time-domain sequence with $e^{j\pi \hat{n}}={\left(-1\right)}^{\hat{n}}$, which leads to the frequency shift block shown in Figure 7d. Accordingly, the final output of the polyphase structure-based decimation is
(17)$$\begin{array}{c} {\hat{y}}_{m}\left(\hat{n}\right)=\alpha e^{j2\pi m\tilde{T}\mu }e^{-j2\pi \hat{n}k_{r}/Q}\times e^{j\pi \hat{n}}=\alpha e^{j2\pi m\tilde{T}\mu }e^{-j2\pi \hat{n}\frac{k_{r}+Q/2}{Q}}. \end{array}$$

### 5.2. Decimation-Based COS (DCOS)

Similar to COS reviewed in Section 3, sensing can also be performed based on ${\hat{y}}_{m}\left(\hat{n}\right)$, leading to DCOS. Taking the two-dimensional DFT of ${\hat{y}}_{m}\left(\hat{n}\right)$ w.r.t. *m* and $\hat{n}$ generates the below RDM (referred to as DCOS-RDM), which has a smaller size than the RDM given in (8) (similarly referred to as COS-RDM),
(18)$$\begin{array}{c} {\hat{Y}}_{b}\left(\hat{k}\right)=\alpha \sum_{\hat{n}=0}^{Q-1}w_{Q}\left(\hat{n}\right)e^{-\frac{j2\pi \hat{n}\left(k_{r}+Q/2\right)}{Q}}e^{-j\frac{2\pi \hat{k}\hat{n}}{Q}}\times \sum_{m=0}^{M-1}w_{M}\left(m\right)e^{j2\pi m\tilde{T}\mu }e^{-j\frac{2\pi bm}{M}}. \end{array}$$

Identifying the peaks of $\left|{\hat{Y}}_{b}\left(\hat{k}\right)\right|$ along $\hat{k}$- and *b*-dimensions can estimate the range and velocity, respectively. Assume that the $\hat{n}$-related summation achieves the maximum at $\hat{k}={\hat{k}}^{†}$. It is easy to see from (18) that the maximum is only achieved when $k_{r}+Q/2+{\hat{k}}^{†}=aQ$, where *a* takes an integer or zero. Solving the equation subject to $k_{r}\in \left[0,Q-1\right]$ yields
(19)$$\begin{array}{c} {\hat{k}}_{r}=Q/2-{\hat{k}}^{†},\;\mathrm{if}\;{\hat{k}}^{†}\in \left[0,Q/2\right];\;{\hat{k}}_{r}=3Q/2-{\hat{k}}^{†},\;\mathrm{if}\;{\hat{k}}^{†}\in \left[Q/2+1,Q-1\right], \end{array}$$
where ${\hat{k}}_{r}$ denotes the estimate of $k_{r}$. Comparing (8) and (18), we see that COS and DCOS have the same velocity measurement. To sum up, DCOS has the following range and velocity estimates, where $\hat{v}$ is given in (9),
(20)$$\begin{array}{c} {\hat{r}}_{d}={\hat{k}}_{r}T_{s}C/2,\;\hat{v}≊b^{†}C/\left(2Mf_{c}\tilde{T}\right). \end{array}$$

Again, we highlight that the illustration in this subsection can be similarly applied to C-COS and the novel sensing framework in Algorithm 1. Details are suppressed here for brevity.

The RDMs of COS and DCOS are compared in Figure 8, where the parameter settings are given in Table 3. From Figure 8a,b, we can see a high similarity between the RDMs of the two methods. This validates the efficacy of the newly introduced decimation. It is noteworthy that DCOS reduces the complexity of generating the RDM shown in the figure by almost an order of magnitude, compared with COS. This can be readily validated by substituting the parameter settings in the above complexity analysis. Figure 8c compares the range cuts between COS and DCOS. We see that DCOS has a slightly wider mainlobe than COS, which is caused by different window lengths. Figure 8d compares the velocity cuts of the two methods. As expected, our design does not affect the velocity measurement.

### 5.3. Comparison between COS and DCOS

Here, we compare COS and DCOS from numerous aspects, through which, the advantages and disadvantages of introducing the efficient decimation are analyzed.

**Computational Complexity:***DCOS reduces the sensing complexity in two ways: first, it has a lower complexity than COS in generating RDM; second, DCOS-RDM has a smaller dimension than COS-RDM, thus reducing the complexity of RDM-dependent target estimation.* From Figure 3, we can see that the computational complexity of COS-RDM is dominated by the 2D-DFT. The complexity is $O\left(MN\log_{2}N+NM\log_{2}M\right)$, which is equal to $O\left(MN\log_{2}\left(MN\right)\right)$ by basic logarithmic laws.

DCOS has two parts of computations: the 2D-DFT for generating DCOS-RDM and decimation. Like COS, the first part of computation has the complexity of $O\left(MQ\log_{2}\left(MQ\right)\right)$. According to Figure 7d, the computational complexity of the polyphase decimation is dominated by the first column of FFTs and the third columns of IFFTs. Their complexity is given by $O(2D\tilde{Q}\log_{2}\tilde{Q})$, since the first (third) column has *D* numbers of $\tilde{Q}$-size FFTs (IFFTs). By designing the anti-aliasing filter such that P≪Q, we can take $\tilde{Q}\approx Q$, and then $O(2D\tilde{Q}\log_{2}\tilde{Q})$ becomes $O\left(2DQ\log_{2}\left(Q\right)\right)$. Note that $2DQ\log_{2}\left(Q\right)$ is much smaller than $MQ\log_{2}\left(MQ\right)$, since *M* can take several hundreds, whereas *D* is around ten. Thus, the computational complexity generating DCOS-RDM is dominated by $O\left(MQ\log_{2}\left(MQ\right)\right)$.

For target detection, COS and DCOS have the same complexity if the same detection algorithm, e.g., CFAR, is used. For target estimation, particularly, the range estimation, COS would have a higher complexity than DCOS. This is because range estimation mainly relies on the row dimension of the RDM, i.e., $Y_{b}\left(k\right)$, given in (8) for COS, and ${\hat{Y}}_{b}\left(k\right)$, given in (18) for DCOS. Whereas ${\hat{Y}}_{b}\left(k\right)$ has *N* rows, ${\hat{Y}}_{b}\left(k\right)$ only has *Q* (a fraction of *N*). A wall-clock time comparison between the complexities of COS and DCOS will be provided shortly through simulations.

**Processing Gain:***COS and DCOS have approximately the same processing gain, which is defined as the difference between the SINR in the RDM, i.e., in (8) and (18), and the SINR in the pre-processed target echo, i.e., in (7).* Let γ denote the SINR of $y_{m}\left(n\right)$ given in (7). Although noises are not explicitly shown in the signal models, the SINR change is easy to track. COS-RDM is obtained from a two-dimensional DFT of $y_{m}\left(n\right)$, and, hence, the SINR in COS-RDM becomes MNγ. Namely, the processing gain of COS is given by MN.

DCOS decimates $y_{m}\left(n\right)$ first. The decimated version ${\hat{y}}_{m}\left(\hat{n}\right)$ given in (17) has the SINR of Dγ, since the decimation with factor *D* does not change the signal power whereas it reduces the noise power by *D* times. The two-dimensional DFT performed in (18) improves the SINR to M(Q−P+1)Dγ≈MNγ, where M(Q−P+1)≈MQ and the approximation is valid given P≪Q. We see that the processing gain of DCOS is approximately MN.

**Remark** **5.**
*The impact of P on DCOS can be non-trivial. For instance, as P increases, a higher quality filter can be obtained (e..g, one with a lower passband ripple, stronger stopband attenuation and narrower transition bandwidth); however, a lower processing gain, as given by M(Q−P+1), is yielded. Analytically, it is difficult to tell which of the following dominates: the SINR improvement earned by a better filter or the SINR degrading caused by the reduced processing gain. To this end, we resort to simulation next.*


Figure 9a,b illustrate that, as *P* increases from 1 to 50, the SINR in DCOS-RDM first increases, then plateaus and then decreases. The same pattern is seen for both small and large values of γ. From this observation, we conclude that the SINR in DCOS-RDM can be maximized by properly setting *P*. For the OFDM system configured in Figure 9, the maximum is achieved at P=16. Using this value, we compare in Figure 9c the SINR in DCOS-RDM with that in COS-RDM as γ increases. We see that the SINRs achieved by COS and DCOS are almost identical in the whole region of γ. Note that the difference between the *y*-axis and *x*-axis is the processing gain. Thus, the results in Figure 9c validate COS and DCOS having approximately the same processing gain.

**Range and Velocity Measurement:***COS and DCOS share the same maximum unambiguous range/velocity, and they also have the same range/velocity resolution.* In terms of velocity, the above statement is because the decimation does not incur any change to Doppler-related information, as manifested in (8) and (18). Based on (8), the range of Doppler frequency that can be unambiguously estimated is $\mu \in [-\frac{1}{2\tilde{T}},\frac{1}{2\tilde{T}}]$, where $\frac{1}{\tilde{T}}$ resembles the sampling frequency along the Doppler dimension. Since the number of samples is *M*, the Doppler frequency resolution is $\Delta_{\mu }=\frac{1}{\tilde{T}M}$. Given the relation μ=2v/λ, we obtain the range of unambiguous velocity, i.e., $v\in [-\frac{\lambda }{4\tilde{T}},\frac{\lambda }{4\tilde{T}}]$, and the velocity resolution, as given by $\Delta_{v}=\frac{\lambda }{2\tilde{T}M}$.

It terms of ranging, we can see from (8) and (18) that the range estimation has turned into a problem of identifying $k_{r}$ in both COS and DCOS. Since $k_{r}\left(=\left\lfloor 2rB/C\right\rceil \right)$ is independent of the sampling rate (or range dimension) in different RDMs, its estimate remains the same for COS and DCOS in theory. As illustrated in Section 3, $k_{r}\leq Q$ is required for OFDM sensing. Let *R* denote the maximum unambiguous detectable range. Solving 2RB/C=Q, we obtain $R=\frac{CQ}{2B}$ for both COS and DCOS. We can see from (8) and (18) that the resolution of $k_{r}$ detection is unit one for both methods, and, hence, the range resolution, denoted by $\Delta_{r}$, can be solved from $2\Delta_{r}B/C=1$, leading to $\Delta_{r}=\frac{C}{2B}$.

**Windowing Effect:***For ranging, COS can achieve a better windowing effect than DCOS in the sense that COS has a narrower range mainlobe than DCOS given the same attenuation of the peak sidelobe, whereas, for the velocity measurement, the two methods have the same windowing effect.* The reason is because the decimation in DCOS reduces the number of samples, and, hence, the window length, along the range dimension (compared with those of COS), whereas the decimation does not affect the velocity dimension.

Figure 10 compares the specific detection and estimation performances of C-COS and DCOS. CFAR is employed to evaluate the detection performance, where a 10×8 rectangular window is used to filter the RDMs (in power). The numbers of guard intervals are two and four in the Doppler and range dimensions, respectively. For range and Doppler estimations, the method in [48] is employed. From Figure 10a–c, we can see that C-COS and DCOS have a very similar detection and estimation performance. This is not surprising, as DCOS, simply removing redundancy through signal decimation, does not lose any essential information related to targets. From Figure 10a, we can see that DCOS has a slightly lower detection probability compared to C-COS. This can be caused by the decimation filter with inevitable pass-band ripples and transition frequency bands. From Figure 10d, we can see clearly that DCOS has a non-trivially lower running time than C-COS, which validates the reduced complexity of DCOS.

## 6. Conclusions

In this paper, we first provide an overview of existing works on OFDM sensing. Through the overview, we highlight some low-complexity sensing methods that have gained great popularity. We then point out some critical issues of the methods that have long been overlooked. To raise the awareness of these issues, we provide a short tutorial, providing the fundamental basis for the sequential review of some recent research efforts in addressing the issues. To further reduce the sensing complexity, we develop a novel method that reduces the dimension of RDM by removing the signal redundancy. Although the recent research efforts, including our own [39,40], have relieved some issues pointed out at the end of Section 1, we are still facing non-trivial challenges using communication waveforms for sensing. Some are highlighted below.

(1)As demonstrated in Figure 6, the sensing framework reviewed in Section 4 can have fake targets. This calls for new methods/designs to either differentiate the fake from true targets or to holistically design the core parameters of the sensing framework to prevent the fake targets from presenting. Moreover, though several core parameters are shown to have a significant impact on the performance of the sensing framework, a scenario-adaptive selection of the parameters is still missing;(2)Most instances of multi-/single-carrier communication-based sensing reviewed in Section 1 and Section 4 are based on single-antenna transceivers. It may not be easy to extend these methods/designs to MIMO communications. Note that signals transmitted from multiple antennas in MIMO communications are not as orthogonal as those in conventional orthogonal MIMO radars. This is more the case when communication signals are subjected to little or no changes. Although the methods reviewed in Section 4 do not require any changes made to a communication transmitter, they alter the signal format at the sensing receiver. Consequently, they can weaken the signal orthogonality among antennas if orthogonal waveforms are employed by the communication transmitter array. Effective sensing using MIMO communication signals needs further investigation;(3)Practical communication systems apply pulse-shaping filters at transmitting and receiving sides. The differences between transceiver filters and other hardware imperfections can have a non-trivial impact on the sensing performance. Such an impact, however, has not been taken into account in the method design and evaluation of most works, including this one. Evaluating such an impact can be difficult, as the hardware error sources may not be easy to model. Prototype-assisted studies may be a better option to investigate the issue.

## Figures and Tables

**Figure 1 sensors-22-01613-f001:**
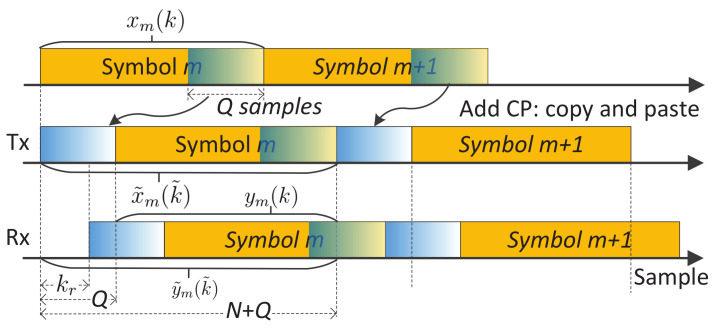
Illustrating the changes in signal timing in OFDM sensing, where CP is short for cyclic prefix and *Q* is the number of samples in a CP. The top signal, $x_{m}\left(k\right)$ given in (1), is the essential part of OFDM symbols. The middle signal, ${\tilde{x}}_{m}\left(\tilde{k}\right)$ given in (3), illustrates the CP-OFDM symbols to be emitted. The bottom signal, ${\tilde{y}}_{m}\left(\tilde{k}\right)$ given in (4), is the baseband echo at the sensing Rx, where the delay of $k_{r}$ samples account for the round-trip traveling from Tx to Rx.

**Figure 2 sensors-22-01613-f002:**
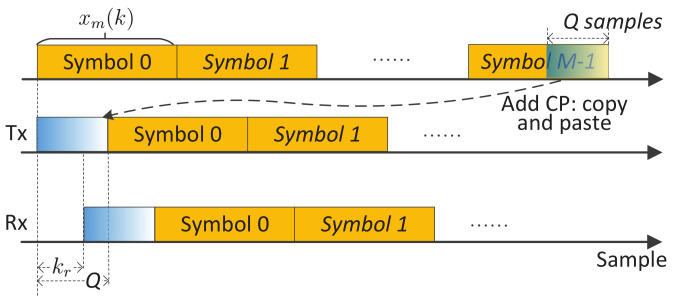
Illustrating the signal timing in RCP-OTFS sensing, where, different from OFDM shown in Figure 1, only a single CP is added to a whole block of symbols.

**Figure 3 sensors-22-01613-f003:**
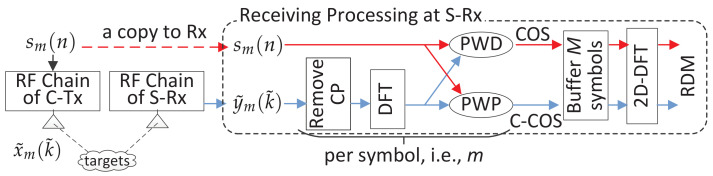
Illustrating the processing diagram of COS and C-COS, where C-Tx stands for communication transmitter, S-Rx for sensing receiver, PWD for point-wise division, PWP for point-wise product and RDM for range–Doppler map.

**Figure 4 sensors-22-01613-f004:**
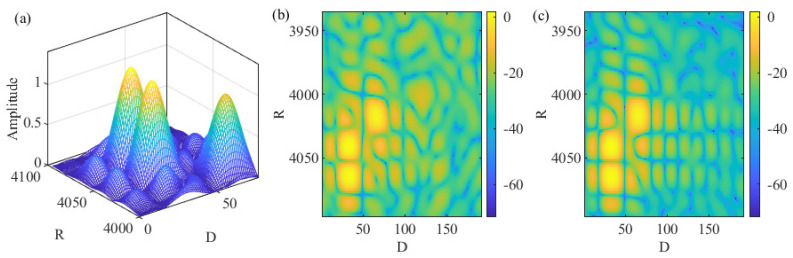
Illustrating RDMs, where “-” in the color bar is the negative sign. Note that $|Y_{m}\left(b\right)|$ with constant modulus $s_{m}\left(n\right)$ is plotted in (**a**), demonstrating OFDM under PSK constellations processed by either PWD or PWP. Moreover, $|Y_{m}\left(b\right)|$ (obtained under PWD) with noise-like $s_{m}\left(n\right)$ is plotted in (**b**). In addition, $|Z_{m}\left(b\right)|$ (using PWP) with noise-like $s_{m}\left(n\right)$ is plotted in (**c**). According to Remark 1, DFT-s-OFDM and OTFS have their frequency-domain signals, i.e., $s_{m}\left(n\right)$, conform to normal distribution. Thus, subfigures (**b**,**c**) can represent either DFT-s-OFDM or OTFS. Here, *R* and *D* stand for range and Doppler grids, respectively. When generating the RDMs as performed in (8) and (11), the DFT sizes in both dimensions are increased by 16 times to make the grids denser.

**Figure 5 sensors-22-01613-f005:**
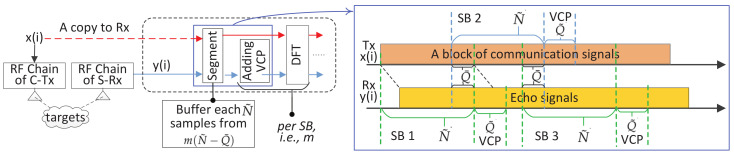
A novel sensing framework that suits OFDM, DFT-s-OFDM and OTFS, where SB stands for sub-block and VCP for virtual CP. The left sub-figure shows the sensing diagram, where the DFT results will go through the last three steps in Figure 3 to generate RDMs. The right sub-figure is a novel signal segmentation proposed in [39], where x(i) can be the middle signal in Figure 1 or Figure 2. *That is, the sensing framework suits OFDM or DFT-s-OFDM with regular CPs (one per symbol), as well as the OTFS with a reduced CP (i.e., a single CP for a long block of symbols)*.

**Figure 6 sensors-22-01613-f006:**
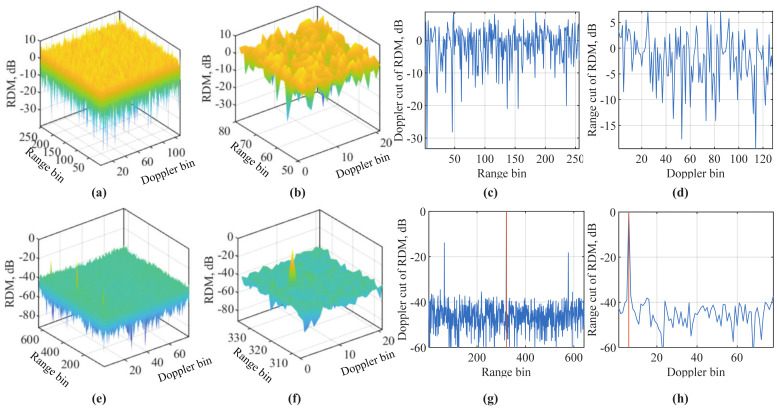
Comparing RDMs of C-COS and the novel sensing framework (NSF) illustrated in Algorithm 1, where simulation parameters are summarized in Table 2, the results in the first row are for C-COS and the results in the second row are for NSF. More specifically, the RDM of C-COS is given in subfigure (**a**), while that of NSF is in subfigure (**e**). Subfigures (**b**) and (**f**) are the zoomed in versions of subfigures (**a**,**e**), respectively, where the zoom-in centers are the true target range and Doppler bins. Subfigures (**c**) and (**d**) illustrate the range and Doppler cuts of the RDM given in subfigure (**a**); similarly, subfigures (**g**) and (**h**) give those of the RDM in (**e**). *Note that COS and NSF are performed with the same communication-transmitted and sensing echo signals*.

**Figure 7 sensors-22-01613-f007:**
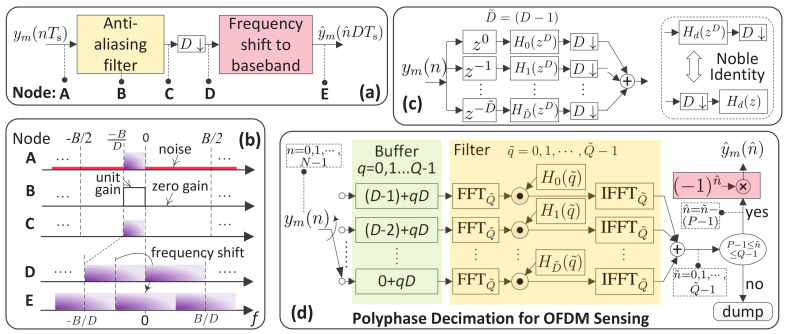
(**a**) Illustration of general steps for decimation; (**b**) spectrum features at different stages of decimation; (**c**) decomposing the anti-aliasing filter in (**a**); (**d**) the polyphase structure-based decimation specifically tailored for OFDM sensing.

**Figure 8 sensors-22-01613-f008:**
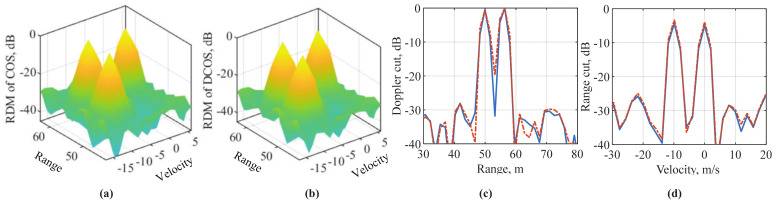
Illustration of target detection, where COS-RDM is given in (**a**), DCOS-RDM in (**b**), the range cuts at v=−10 m/s are shown in (**c**) and the velocity cuts at r=56 m in (**d**). Most settings in Table 3 are again used here, except that the number of OFDM symbols is M=256 and the hamming window is used in (8) and (18) for both range and velocity measurements. In addition, three targets are set here. Their ranges and velocities are [50,56,56] m and [−10,−10,0] m/s, respectively. Note that the symbol “-” in the axes of all subfigures is the minus sign (not hyphen).

**Figure 9 sensors-22-01613-f009:**
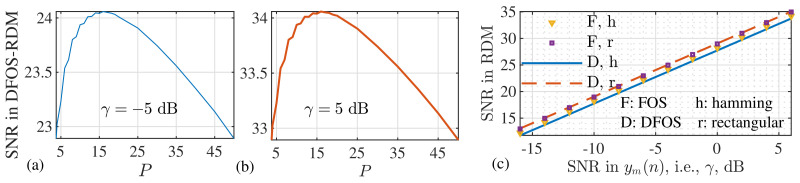
Illustration of SNR in DCOS-RDM versus *P* in (**a**,**b**); and (**c**) a comparative illustration of the SNR in RDM of both COS and DCOS versus γ, the SNR in (7). Parameter settings are summarized in Table 3.

**Figure 10 sensors-22-01613-f010:**
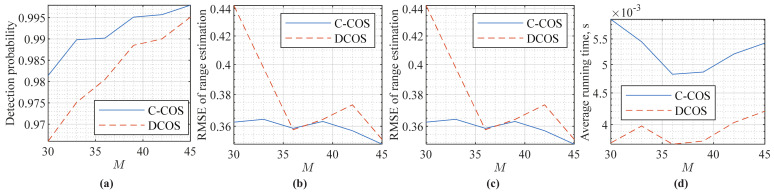
Comparing C-COS and DCOS in terms of detection and estimation performances, where the OFDM parameters are given in Table 3, and a single unit-power target is set here with range and velocity randomly generated over $10^{4}$ independent trials. (**a**) illustrates the detection probability of the two methods under $10^{-4}$ false-alarm rate and γ=−60 dB. (**b**,**c**) illustrates the range and velocity estimation performance, respectively, where the estimation method [48] is employed for both parameters. (**d**) compares the wall-clock time per run, including RDM generation, detection and estimation, for the two methods, as averaged over $10^{4}$ trials.

**Table 1 sensors-22-01613-t001:** Target parameters, where three targets are simulated, $U_{\left[x,y\right]}$ denotes the uniform distribution in [x,y] and γ denotes the SNR ^1^.

Var	Value	Var	Value
α	$\left[e^{jx_{1}},e^{jx_{2}},e^{jx_{3}}\right]\;(x_{i}$∼$U_{\left[0,2\pi \right]}\;\forall i)$	$k_{r}$	[2,3.5,5]
$\mu \tilde{T}M$	[2,2,4]	*M*	128
*N*	256	γ	−10 dB

^1^ Note that *γ* is defined based on the time-domain echo signal given in (4). The signal power is averaged over the three targets and, hence, is one. The noise, though not shown in (4), is a complex Gaussian signal with the power set as 10 dB in the simulation.

**Table 2 sensors-22-01613-t002:** Simulation settings for Figure 6, where $f_{c}$ is the carrier frequency, $f_{s}$ denotes the sampling frequency and γ is the SNR of the time-domain echo signal y(i).

Var	Value	Var	Value
α	$e^{jx}\;(x_{i}$∼$U_{\left[0,2\pi \right]}\;\forall i)$	$k_{r}$	320
μ	480 Hz	*M*	128
*N*	256	*Q*	64
$f_{c}$	2.4 GHz	$f_{s}$(=*B*)	3.84 MHz
γ	−10 dB	–	–

**Table 3 sensors-22-01613-t003:** Simulation settings for comparing COS and DCOS, which are with reference to [12] [Tab.2].

Var	Value	Var	Value
α	$e^{jx_{i}}\;(x_{i}$∼$U_{\left[0,2\pi \right]},\;\forall i)$	*M*	1
*N*	1024	*Q*	128
$f_{c}$	24 GHz	$f_{s}\left(=B\right)$	93 MHz

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
