# Peer review of "Joint Communications and Sensing Employing Multi- or Single-Carrier OFDM Communication Signals: A Tutorial on Sensing Methods, Recent Progress and a Novel Design"

_sensors, 2022, doi:10.3390/s22041613_

Round 1

Reviewer 1 Report

The paper presents an analysis of different versions of OFDM in terms or range-Doppler processing. The first 10 pages have more of a tutorial flavor, while the remaining 7 pages provide more novel research results. I found the work relatively interesting, though the paper can be improved in terms of both presentation and clarity. My specific comment are listed below:

  • A more detailed evaluation similar to Fig. 4 for the different waveforms OFDM, OTFS, and DFTS-OFDM with PWM and PMD would be very interesting to understand then benefits and drawbacks of each approach. Possibly different modulation formats could be considered as well.
  • The Doppler-induced inter-carrier interference (ICI) is neglected. For larger Dopplers, this means that the hop-and-stop model becomes invalid. Please explain under which conditions (for which Dopplers) ICI should be included in the model. 
  • The CP-limited range is just a side-effect of OFDM processing. If the signal is processed in the time-domain, the maximum range is effectively unbounded, since OFDM is just like any other signal. This could be clarified before section 3. 
  • In Section 3, a method proposed by the same authors in [40] is presented. I would appreciate a bit more detail, including a more precise Table 3 (the notation now is very sloppy), a more clear description of the synchronization assumptions (since Fig. 5 operates on a stream rather than sequences of OFDM symbols, the synchronization process should be explained). For instance, is x(i) a sample or a vector? How do x(i) and y(i) relate? Can the sub-blocks be chosen arbitrarily or should there be some relation between the blocks x_m(n) and y_m(n)? Please show images of the 2D-FFTs mentioned at line 5. How are the targets detections from different SBs combined? These details would help the reader understand the operation and benefits of the proposed method from [40]. A comparison with the images from Fig. 4 under the same conditions should also be included.
  • Section 4 provides a low-complexity method to detect targets. There are many such approaches (e.g., first do a coarse 2DFFT to find peaks and then perform another set 2DFFT around the peaks) to reduce complexity. I would like to see a complexity evaluation (wall time) between different such alternatives. Please also clarify if the method is applicable to single-target or also multi-target scenarios. 
  • In the results: it was impossible for me to understand the specific simulation parameters. Please list all used parameters (power, bandwidth, subcarrier spacing, integration time, modulation, target model, noise power, etc), so that the results can be independently verified and reproduced. If possible, relate the selected parameters to references and/or standards.
  • In the results: please provide also range-Doppler maps (and range / Doppler cuts if meaningful) as well as figures of the detection probability (for a certain false alarm rate) and RMSE in range and Doppler vs SNR. These are needed to verify that the method works as intended (and can indeed detect and estimate a target) and that it is better / less complex than standard methods. 

Minor comments:

  • Some issue with section numbering and cross-referencing.
  • Several typos (e.g. sever vs severe).

Author Response

We thank this reviewer very much for his/her time and effort in reviewing our paper. Your valuable comments have all been addressed in the rerevised paper. Please refer to the uploaded response letter for the details. We hope you find our responses and revisions satisfactory. 

Reviewer 2 Report

Minor changes / Typos:

Please try to avoid the use of "we" in the abstract and in the conclusions.
It is better to use impersonal forms...
For example:
we develop a novel method -> a novel method is developed

Line 17:
in both both -> in both

Line 260:
With targets detected -> With detected targets

Line 365:
With this fact noticed -> With this noticed fact 

Line 415:
with transients removed -> with removed transients

General comments:

To understand better the paper and the tutorial for non expert readers, I suggest to add more figures and graphics/plots, if possible.

Author Response

We thank this reviewer very much for his/her time and effort in reviewing our paper. We have now revised the paper, addressing all your comments. Please refer to the uploaded response letter for the details. We hope you find our responses and revisions satisfactory. 

Reviewer 3 Report

This paper develops a novel method to further reduce the sensing complexity. It gives the wide applicability of orthogonal frequency-division multiplexing (OFDM) in modern communications, OFDM sensing has become one of the major research topics of JCAS.

However, I have some questions listed as follows:

  1. The paper is not well organized, in which the novelty and contributions are not very clear in the text. It is too tedious and tedious without stressing the main points of the contributions. Section 2 and section 3 are not the contribution of yourself, it should be simplified.
  2. Some symbols and number in this paper are wrong and unclear, for example the sub-title “0. Background and Motivation”, the number should be started at “1” not “0”.
  3. Is it better to briefly explain the proposed of this paper in the abstract?
  4. The description of decimation-based COS in 4.2 of this paper is not clear.
  5. The comparison of C-COS with COS and DCOS is missing in 4.3 of this paper.
  6. Table 3 is an algorithm, it should be performed in an algorithm manner, not just a table.

Author Response

(The authors gave the same response as above.)

Reviewer 4 Report

-The paper presents an overview of OFDM (Orthogonal Frequency-Division Multiplexing) sensing and a tutorial. The paper also presents a method to reduce sensing complexity over one of the most efficient methods to date.
-The technical aspects of the research presented seem sound and detailed.
-While the paper presents an overview of OFDM sensing and a tutorial, I think you should add more content to the "Conclusions" section as well as mention future work.
-There are some typos and grammatical errors in the text that should be checked.

Author Response

(The authors gave the same response as above.)

Round 2

Reviewer 1 Report

The authors have addressed all my concerns. Thank you.

Reviewer 3 Report

In the revision, the authors well addressed my concerns raised in the first round of review process.  Therefore I would like to recommend accepting this paper